# P2P: Tuning Pre-trained Image Models for Point Cloud Analysis with Point-to-Pixel Prompting

**Ziyi Wang**[*]  **Xumin Yu**[*]  **Yongming Rao**[*]
**Jie Zhou**    **Jiwen Lu**[†]
Department of Automation, Tsinghua University, China
`{wziyi22, yuxm20}@mails.tsinghua.edu.cn;`
`raoyongming95@gmail.com;`
`{lujiwen, jzhou}@tsinghua.edu.cn`

## Abstract

Nowadays, pre-training big models on large-scale datasets has become a crucial topic in deep learning. The pre-trained models with high representation ability and transferability achieve a great success and dominate many downstream tasks in natural language processing and 2D vision. However, it is non-trivial to promote such a pretraining-tuning paradigm to the 3D vision, given the limited training data that are relatively inconvenient to collect. In this paper, we provide a new perspective of leveraging pre-trained 2D knowledge in 3D domain to tackle this problem, tuning pre-trained image models with the novel *Point-to-Pixel prompting* for point cloud analysis at a minor parameter cost. Following the principle of prompting engineering, we transform point clouds into colorful images with geometry-preserved projection and geometry-aware coloring to adapt to pre-trained image models, whose weights are kept frozen during the end-to-end optimization of point cloud analysis tasks. We conduct extensive experiments to demonstrate that cooperating with our proposed Point-to-Pixel Prompting, better pre-trained image model will lead to consistently better performance in 3D vision. Enjoying prosperous development from image pre-training field, our method attains 89.3% accuracy on the hardest setting of ScanObjectNN, surpassing conventional point cloud models with much fewer trainable parameters. Our framework also exhibits very competitive performance on ModelNet classification and ShapeNet Part Segmentation. Code is available at https://github.com/wangzy22/P2P.

## 1   Introduction

With the rapid development of deep learning and computing hardware, neural networks are experiencing explosive growth in model size and representation capacity. Nowadays, pre-training big models has become an important research topic in both natural language processing [14, 52, 4, 60] and computer vision [51, 53, 20, 57], and has achieved a great success when transferred to downstream tasks with fine-tuning [21, 10, 9, 2, 20] or prompt-tuning [51, 45, 67, 59, 29, 35] strategies. Fine-tuning is a traditional tuning strategy that requires a large amount of trainable parameters, while prompt tuning is a recently emerged lightweight scheme to convert downstream tasks into the similar form as the pre-training task. However, such prevalence of the pretraining-tuning pipeline cannot be obtained without the support of numerous training data in pre-training stage. Language pre-training leading work Megatron-Turing NLG [60] with 530 billion parameters is trained on 15 datasets containing over 338 billion tokens, while Vision MoE [57] with 14.7 billion parameters is trained on JFT-300M dataset [62] including 305 million training images.

---

[*]Equal contribution.   [†]Corresponding author.

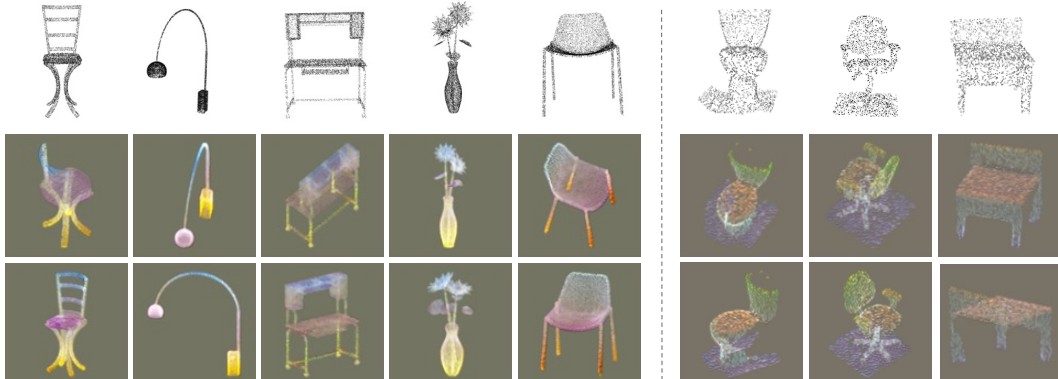

Figure 1: **Images produced by our Point-to-Pixel Prompting.** We show the original point clouds (top line) and the projected colorful images produced by our P2P of synthetic objects from ModelNet40 (left five columns) and real-world objects from ScanObjectNN (right three columns) from two different projection views.

Unfortunately, the aforementioned convention of pre-training big models on large-scale datasets and tuning on downstream tasks has encountered obstacles in 3D vision. 3D visual perception is gaining more and more attention given its superiority in many emerging research fields including autonomous driving [28, 80], robotics vision [8, 78] and virtual reality [41, 71]. However, obtaining abundant 3D data such as point clouds from LiDAR is neither convenient nor inexpensive. For example, the widely used object-level point cloud dataset ShapeNet [7] only contains 50 thousand synthetic samples. Therefore, pre-training fundamental 3D models with limited data remains an open question. There are some previous literature [77, 68, 81] that attempts to develop specific pre-training strategies on point clouds with limited training data, such as Point Contrast [77], OcCo [68] and Point-BERT [81]. Although they prove that the pretraining-finetuning pipeline also works well in the 3D domain, the imbalance between numerous trainable parameters and limited training data may lead to insufficient optimization or overfitting problems.

Different from the previous methods that directly pre-train models on 3D data, we propose to transfer the pre-trained knowledge from 2D domain to 3D domain with appropriate prompting engineering, since images and point clouds display the same visual world and share some common knowledge. In this way, we address the data-starvation problem in the 3D domain, given that the pre-training strategy is well-studied in the 2D field with abundant training data and that prompt-tuning on 3D tasks does not require much 3D training data. To the best of our knowledge, we are the first work to transfer knowledge in pre-trained image models to 3D vision with a novel prompting approach. More specifically, we propose an innovative Point-to-Pixel Prompting mechanism that transforms point clouds into colorful images with geometry-preserved projection and geometry-aware coloring. Examples of produced colorful images are shown in Figure 1. Then the colorful images are fed into the pre-trained image model with frozen weights to extract representative features, which are further deployed to downstream task-specific heads. The conversion from point clouds to colorful images and the end-to-end optimization pipeline promote the bidirectional knowledge flow between points and pixels. The geometric information from point clouds is mostly retained in projected images via our geometry-preserved projection, while the color information of natural images from the pre-trained image model is transmitted back to colorless point clouds via the cooperation between the geometry-aware coloring module and the fixed pre-trained image model.

We conduct extensive experiments to demonstrate that with our Point-to-Pixel Prompting, enlarging the scale of the same image model will result in higher point cloud classification performance, which is consistent with the observations in image classification. This suggests that we can take advantage of the successful researches in pre-training big image model, opening up a new avenue for point cloud analysis. With much fewer trainable parameters, we achieve comparable results with the best object classification methods on both synthetic ModelNet40 [74] and real-world ScanObjectNN [65]. We also demonstrate the potential of our method to perform dense predictions like part segmentation on ShapeNetPart [79]. In conclusion, our Point-to-Pixel Prompting (P2P) framework explores the feasibility and ascendancy of transferring image pre-trained knowledge to the point cloud domain, promoting a new pre-training paradigm in 3D point cloud analysis.

## 2 Related Work

### 2.1 Visual Pre-training

Pre-training visual models has been studied thoroughly in the image domain. Supervised pre-training [15, 82, 6] on classification task with large-scale dataset is a traditional practice and is stimulated by the boosting development of the ever-growing fundamental vision models [22, 23, 15, 37]. Weakly-supervised pre-training methods [63, 3, 76, 46] use less annotations while unsupervised pre-training approaches [21, 10, 9, 2, 20, 17] introduces no task-related bias and brings higher transferability to various downstream tasks.

Different from the prosperity of pre-training image models, pre-training 3D models is still under development. Many researches have developed self-supervised learning mechanisms with various pretext tasks such as solving jigsaw puzzles [58], orientation estimation [47], and deformation reconstruction [1]. Inspired by pre-training strategies in image domain, Point Contrast [77] adopts contrastive learning principle while OcCo [68], Point-BERT [81] and Point-M2AE [83] introduce reconstruction pretext tasks for better representation learning. However, the data limitation in 3D domain remains a large obstacle in developing better pre-training strategies.

### 2.2 Prompt Tuning

Prompt tuning is an important mechanism whose principle is to adapt downstream tasks with limited annotated data to the original pre-training task at a minimum cost, thus exploiting the pre-trained knowledge to solve downstream problems. It is first proposed in the natural language processing community [33], and has been leveraged in many vision-language models. At first, hand-crafted prompting methods [45, 4] are promoted and their followers [67, 59] develop an automated searching algorithm to select discrete prompt tokens within a large corpus. Recently, continuous prompting methods [31, 29, 35, 34] are becoming the mainstream given their flexibility and high performance.

On the contrary, the development in prompting visual pre-training models lags behind. L2P [70] proposes a prompt pool for continual learning problem while VPT [24] first introduces continuous prompt tuning framework inspired by P-Tuning [35, 34]. As far as we are concerned, there is no previous work like this paper to discuss tuning pre-trained image models for point cloud analysis with an appropriate prompting mechanism.

### 2.3 Object-level Point Cloud Analysis

Given the unordered data structure of point clouds, early literature has developed voxel-based and point-based methods to construct structural representations for point cloud object analysis. Voxel-based methods [43, 27, 56] partition the 3D space into ordered voxels and perform 3D convolutions for feature extraction. Point-based methods [48, 49, 42, 12, 32, 73, 64, 69, 30] directly process unordered points and introduce various approaches to aggregate local information. Recently, attention-based Transformer [66, 18, 85] architecture has prevailed over other frameworks in vision community and achieved competitive performance in point cloud object analysis.

Besides the aforementioned methods that perform representation learning in the 3D space, there are projection-based methods[61, 25, 72, 19, 16, 84] that leverage multi-view images to represent 3D objects. Recently, MVTN [19] introduces the differentiable rendering technique to build an end-to-end learning pipeline, rendering images online and regressing the optimal projection view. Different from theirs, our work designs a novel prompting engineering scheme, utilizing 2D color knowledge from pre-trained image models that is absent in colorless point clouds. Moreover, our framework is implemented in a faster single-view pattern, as we only select one random projection view during training and don't develop any aggregation strategy to explicitly fuse multi-view knowledge.

## 3 Approach

### 3.1 Overview

The overall framework of our P2P framework is illustrated in Figure 2. The network architecture consists of four components: 1) a geometry encoder to extract point-level geometric features from

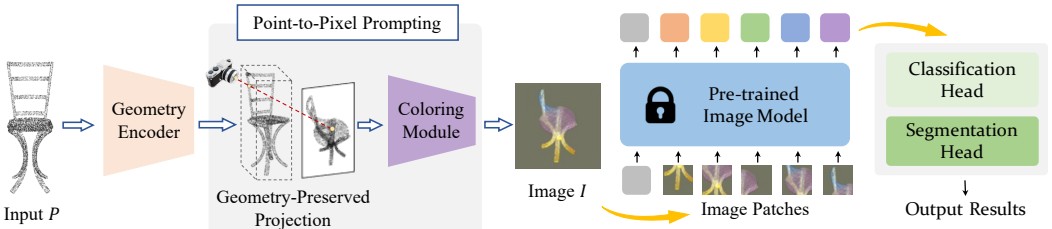

Figure 2: **The pipeline of our proposed P2P framework.** Taking a point cloud $P$ as the input, we first encode the geometry information for each point. Then we sample a projection view and rearrange the point-wise features into an image-style layout to obtain the pixel-wise features with *Geometry-preserved Projection*. The colorless projection will be enriched to produce a colorful image $I$ with the color information via a learnable *Coloring Module*. Our P2P framework can be easily transferred to several downstream tasks with a task-specific head with the help of the transferable visual knowledge from the pre-trained image model. We take the classical Vision Transformer [15] as our pre-trained image model for illustration in this pipeline.

the input point clouds, 2) a Point-to-Pixel Prompting module to produce colorful images based on geometric features, 3) a pre-trained image model to leverage pre-trained knowledge from image domain, and 4) a task-specific head to perform various kinds of point cloud tasks. We will introduce the geometry encoder, the Point-to-Pixel Prompting module and task-specific heads in detail in the following sections. As for the choice of the pre-trained image model, we investigate both convolution-based and attention-based architectures in Section 4.2.1.

With the proposed architecture that can be optimized in an end-to-end manner, we are able to exploit 2D pre-trained knowledge for point cloud analysis from two perspectives. In the forward process, the point clouds are projected into images with preserved geometry information and the resulting images can be recognized and handled by the pre-trained image model. In the backward optimization, the frozen pre-trained weights of the image model act as an anchor and guide the learnable coloring module to learn extra color knowledge for colorless point clouds, without explicit manual interference and only under the indirect supervision from the overall target functions of downstream tasks. Therefore, the resulting colorful images are expected to mimic patterns in 2D images and to be distinguishable for the pre-trained image model in downstream tasks.

## 3.2 Point Cloud Feature Encoding

One of the most significant advantages of 3D point clouds over 2D images is that point clouds contain more spatial and geometric information that is compressed or even lost in flat images. Therefore, we first extract geometry features from point clouds for better spatial comprehension, implementing a lightweight DGCNN [69] to extract local features of each point.

Given an input point cloud $P \in \mathbb{R}^{N \times 3}$ with $N$ points, we first locate $k$-nearest neighbors $\mathcal{N} \in \mathbb{R}^{N \times k \times 3}$ of each point. Then for each local region, we implement a small neural network $h_\Theta$ to encode the relative position relations between the central point $p_i$ and the local neighbor points $\mathcal{N}_{p_i}$. Then we can obtain geometric features $F = \{f_i, 0 \leq i < N\} \in \mathbb{R}^{N \times C}$ with dimension $C$:

$$f_i = \mathrm{maxpool}_{\mathcal{N}_{p_i}}(h_\Theta(\mathrm{concat}_{\mathcal{N}_{p_i}}(\mathrm{x}_i, \mathrm{x}_j - \mathrm{x}_i))), \tag{1}$$

where $\mathrm{x}_i, \mathrm{x}_j$ are coordinates of $p_i, p_j$ respectively, $\mathrm{maxpool}_{\mathcal{N}_{p_i}}$ and $\mathrm{concat}_{\mathcal{N}_{p_i}}$ stand for max-pooling and concatenation within all points $p_j$ in local neighbor region $\mathcal{N}_{p_i}$ respectively.

## 3.3 Point-to-Pixel Prompting

Following the principle of prompt tuning mechanism introduced in Section 2.2, we propose Point-to-Pixel Prompting to adapt point cloud analysis to image representation learning, on which the image model is initially pre-trained. As illustrated in Figure 2, we first introduce geometry-preserved projection to transform the 3D point cloud into 2D images, rearranging 3D geometric features according to the projection correspondences. Then we propose a geometry-aware coloring module to dye projected images, transferring 2D color knowledge in the pre-trained image model to the colorless point cloud and obtaining more distinguishable images that can be better recognized by the pre-trained image model.

### 3.3.1 Geometry-Preserved Projection

Once obtaining geometric features $F \in \mathbb{R}^{N \times C}$ of the input point cloud $P$, we further rearrange them into an image-style layout $\hat{F} \in \mathbb{R}^{H \times W \times C}$ to prepare for producing colorful image $I$, where $H, W$ are height and width of the target image. We elaborately design a geometry-preserved projection to avoid information loss when casting 3D point clouds to 2D images.

The first step is to find spatial correspondence between point coordinates $\mathrm{X} \in \mathbb{R}^{N \times 3}$ and image pixel coordinates $\mathrm{Y} \in \mathbb{R}^{N \times 2}$. Since there is a dimensional diminishing during the projection process, we randomly select a projection view during training to construct a stereoscopic space with flat image components. Equivalently, we rotate the input point cloud with rotation matrix $R \in \mathbb{R}^{3 \times 3}$ to get 3D coordinates $\tilde{\mathrm{X}}$ after rotation: $\tilde{\mathrm{X}} = \mathrm{X}R^T$. The rotation matrix $R$ is constructed through two steps: first rotating around the axis $u_\theta = (0, 0, 1)$ by angle $\theta$, then rotating around the axis $u_\phi = (\sin\theta, -\cos\theta, 0)$ by angle $\phi$, where $\theta \in [-\pi, \pi]$ and $\phi \in [-\pi/2, \pi/2]$ are random rotation angles during training and fix-selected angles during inference. Then we just omit the final dimension $\tilde{\mathrm{X}}_{:,2}$ and evenly split the first two dimensions into 2D grids: $y_{i,d} = \lfloor \tilde{x}_{i,d}/g_d \rfloor$, where $0 \leq i < N$ denotes point index, $d = 0, 1$ denotes coordinate dimension, $g_d$ denotes grid size at dimension $d$.

The second step is to rearrange per-point geometric features $F$ into per-pixel $\hat{F}$ according to coordinates correspondence between X and Y. If there are multiple points $\mathcal{S}_{h,w} = \{p_j\}$ falling in the same pixel at $(h, w)$, which is a common situation, we add the features of these points altogether to produce the pixel-level feature: $\hat{f}_{h,w} = \sum_{p_j \in \mathcal{S}_{h,w}} f_j$. The summation operation brings two advantages related to geometry-preserved design. On the one hand, we consider all points in one pixel instead of keeping the foremost point according to depth and occlusion relation. Therefore, we are able to represent and optimize all points in one image and produce images containing semitransparent objects with richer geometric information as shown in Figure 1. On the other hand, we conduct a summation operation instead of taking the average, resulting in larger feature values when there are more points in one pixel. Such design maintains the spatial density information of point clouds during the projection process, which is lacked in image representations and is critical in preserving geometry knowledge.

In conclusion, the geometry-preserved projection produces geometry-aware image features that contain plentiful spatial knowledge of the object. Note that we only use one projection view during training and do not explicitly design any aggregation functions for multi-view feature fusion. Therefore, we follow a more efficient single-view projection pipeline than its multi-view counterpart.

### 3.3.2 Geometry-Aware Coloring

Despite that 3D point cloud contains richer geometric knowledge than 2D images, colorful pictures embrace more texture and color information than colorless point clouds, which is also decisive in visual comprehension. The frozen image model pre-trained on abundant images learns to perceive the visual world not only based on object shape and outlines, but also heavily relied on discriminative colors and textures. Therefore, the image feature map $\hat{F}$ that contains only geometric knowledge and lacks color information is not most suitable for the pre-trained image model to understand and process. In order to better leverage pre-trained 2D knowledge of the frozen image model, we propose to predict colors for each pixel, explicitly encouraging the network to migrate color knowledge in the pre-trained image model to $\hat{F}$ via the end-to-end optimization. Since the input $\hat{F}$ contains rich geometry information that will heavily affect the coloring process, the resulting images are expected to display different colors on different geometry parts, which has been verified in Figure 1.

More specifically, we design a lightweight 2D neural network $g_\Phi$ to predict RGB colors $\mathrm{C} = \{c_{h,w}\} \in \mathbb{R}^{H \times W \times 3}$ for each pixel $(h, w)$: $c_{h,w} = g_\Phi(\hat{f}_{h,w})$. We implement several $3 \times 3$ convolutions in $g_\Phi$ for image smoothing, as the initial projected image feature $\hat{F}$ are relatively discontinuous due to the sparsity of the original point cloud. Therefore, the smoothing operation is critical in producing more realistic images that the pre-trained image model can recognize. The resulting colorful images are then prepared for further image-level feature extraction through the pre-trained image model.

### 3.4 Optimization on Downstream Tasks

Take ViT as the pre-trained image model for example. The outputs from the pre-trained image model are image token features $\bar{F} \in \mathbb{R}^{N_t \times C_t}$ and one class token feature $\bar{f}_{\mathrm{cls}} \in \mathbb{R}^{1 \times C_t}$, where $N_t$ is the

number of image patches and $C_t$ is the token feature dimension. For different downstream tasks, we design different task-specific heads and optimization strategies.

**Object Classification** For object classification, we follow the common protocol in image Transformer models to utilize the class token $\bar{f}_{\text{cls}}$ as the input to the classifier CLS implemented as only one linear layer: $p = \text{softmax}(\text{CLS}(\bar{f}_{\text{cls}}))$. We use the CrossEntropy loss as the optimization target.

**Part Segmentation** For part segmentation, we rearrange the token features $\bar{F}$ into image layouts and upsample them to $H \times W$. Then we design a lightweight 2D segmentation head SEG based on SemanticFPN [26] or UPerNet [75] to predict per-pixel segmentation logits: $p_{h,w} = \text{softmax}(\text{SEG}(\bar{f}_{\text{h,w}}))$. Given that multiple points may correspond to one pixel and that we train the network in a single view pattern, projecting per-pixel predictions back to 3D points will cause supervision conflict. Instead, we project 3D labels into 2D image-style labels, exactly as how the point cloud is projected. Then we implement a per-pixel multi-label CE loss as there may be points from multiple classes projected to the same pixel: $\mathcal{L}_{\text{seg}} = \sum_{h,w} \sum_k -y_{h,w,k} \log p_{h,w,k}$. The values of multi-hot 2D label $y$ are assigned according to projection correspondences, satisfying $\sum_k y_{h,w,k} = 1$. Supervision in 2D domain speeds up the training procedure without much information loss, since we keep all features of points in one pixel and the optimization target is accordingly based on their category distributions. During inference, we select multiple projection views and re-project 2D per-pixel segmentation results back to 3D points, fusing multi-view predictions. Therefore, the per-point segmentation is decided by the most evident predictions from the most distinguishable projection directions.

## 4 Experiments

### 4.1 Datasets and Experiment Settings

**Datasets.** We conduct classification on ModelNet40 [74] and ScanObjectNN [65], while ShapeNet-Part [65] is utilized for part segmentation. **ModelNet40** is a synthetic 3D dataset containing 12,311 CAD models from 40 categories. **ScanObjectNN** samples from real-world scans with background and occlusions. It contains 2,902 samples from 15 categories, and we conduct experiments on the perturbed (PB-T50-RS) variant. **ShapeNetPart** samples 16,881 objects covering 16 shape categories from the synthetic ShapeNet and annotates each object with part-level labels from 50 classes.

**Implementation Details.** We utilize AdamW [40] optimizer and CosineAnnealing scheduler [39], with learning rate $5e^{-4}$ and weight decay $5e^{-2}$. We freeze the weights of the pre-trained image model except for normalization layers. The model is trained for 300 epochs with a batch size of 64. During training, the rotation angle $\theta, \phi$ are randomly selected from $[-\pi, \pi]$ and $[-0.4\pi, -0.2\pi]$ to keep the objects standing upright. During inference, we evenly choose 10 values of $\theta$ and 4 values of $\phi$ to produce 40 views for majority voting. Please refer to the supplementary for architectural details.

### 4.2 Object Classification

#### 4.2.1 Results

**Main Results.** We implement our P2P framework with different image models of different scales, ranging from convolution-based ResNet [22] and ConvNeXt [38] to attention-based Vision Transformer [15] and Swin Transformer [36]. These image models are pre-trained on ImageNet-1k [13] with supervised classification. We report the image classification performance of the original image model, the number of trainable parameters after Point-to-Pixel Prompting, and the classification accuracy on ModelNet40 and ScanObjectNN datasets, as shown in Table 1.

From the quantitative results and accuracy curve, we can conclude that enlarging the scale of the same image model will result in higher classification performance, which is consistent with the observations in image classification. Therefore, our proposed P2P prompting can benefit 3D domain tasks by leveraging the prosperous development of 2D visual domain, including abundant training data, various pre-training strategies and superior fundamental architectures.

**Comparisons with Previous Methods.** Comparisons with previous methods on the ModelNet40 and ScanobjectNN are shown in Table 2. For baseline comparisons, we select methods [49, 64, 69, 42,

Table 1: **Classification results on ModelNet40 and ScanObjectNN.** For different image models, we report the image classification performance (IN Acc.) on ImageNet-1k [13] dataset. After migrating them to point cloud analysis with Point-to-Pixel Prompting, we report the number of trainable parameters (Tr. Param.), performance on ModelNet40 dataset (MN Acc.) and performance on ScanObjectNN dataset (SN Acc.).

(a) ResNet [22].

| Image Model | IN Acc. | Tr. Param. | MN Acc. | SN Acc. |
|---|---|---|---|---|
| ResNet-18 | 69.8 | 109 K | 91.6 | 82.6 |
| ResNet-50 | 76.1 | 206 K | 92.5 | 85.8 |
| ResNet-101 | 77.4 | 257 K | 93.1 | 87.4 |

(b) Vision Transformer [15].

| Image Model | IN Acc. | Tr. Param. | MN Acc. | SN Acc. |
|---|---|---|---|---|
| ViT-T | 72.2 | 99 K | 91.5 | 79.7 |
| ViT-S | 79.8 | 116 K | 91.8 | 81.6 |
| ViT-B | 81.8 | 150 K | 92.7 | 83.4 |

(c) Swin Transformer [37].

| Image Model | IN Acc. | Tr. Param. | MN Acc. | SN Acc. |
|---|---|---|---|---|
| Swin-T | 81.3 | 136 K | 92.1 | 82.9 |
| Swin-S | 83.0 | 154 K | 92.5 | 83.8 |
| Swin-B | 83.5 | 178 K | 92.6 | 84.6 |

(d) ConvNeXt [38].

| Image Model | IN Acc. | Tr. Param. | MN Acc. | SN Acc. |
|---|---|---|---|---|
| ConvNeXt-T | 82.1 | 126 K | 92.6 | 84.9 |
| ConvNeXt-S | 83.1 | 140 K | 92.8 | 85.3 |
| ConvNeXt-B | 83.8 | 159 K | 93.0 | 85.7 |
| ConvNeXt-L | 84.3 | 198 K | 93.2 | 86.2 |

(e) Accuracy on point cloud classification datasets vs. ImageNet-val for different models.

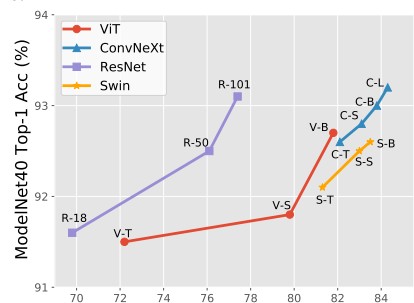

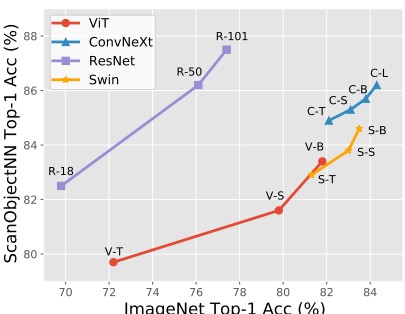

19, 54, 50] that focus on developing 3D architectures and do not involve any pre-training strategies. We also select traditional pre-training work [81, 68, 44] in 3D domain. For our P2P framework, we show results of two versions: (1) baseline version with ResNet-101 as the image model, (2) advanced version with HorNet-L [55] pre-trained on ImageNet-22k dataset [13] as the image model, additionally replacing the linear head with a multi-layer perceptron (MLP) as the classifier.

From the results we can draw three conclusions. Firstly, P2P outperforms traditional 3D pre-training methods. This suggests that the pre-trained knowledge from 2D domain is useful for solving 3D recognition problems and is better than directly pre-training on 3D datasets with limited data. Secondly, we achieve state-of-the-art performance on ScanObjectNN. Therefore, our P2P framework fully exploits the potential of pre-training knowledge from image domain and opens a new avenue for point cloud analysis. Finally, P2P performs relatively better on real-world ScanObjectNN than synthetic ModelNet. This may be caused by the data distribution of ScanObjectNN being more similar to the pre-trained ImageNet dataset, as they both contain visualizations of objects from the natural world. This prosperity reveals the potential of P2P in real-world applications.

**Visualization Analysis.** The visualizations of our projected colorful images are shown in Figure 1. The first line shows point cloud samples, the second and third lines illustrate the colorful images from different projection views. Our geometry-preserved projection design maintains most spatial information, resulting in images of semitransparent objects that avoid occlusion problems, such as the chair leg in the second row $5^{th}$ column.

### 4.2.2 Ablation Studies

To investigate the architecture design and training strategy of our proposed framework, we conduct extensive ablation studies on ModelNet40 classification. Except for further notice, we use the base version of Vision Transformer (ViT-B-1k) that is pre-trained on ImageNet-1k dataset as our image model. Illustrations of our ablation settings can be found in Figure 3.

Table 2: **Comparisons on classification accuracy (Acc.) with previous literature on point cloud datasets.**
We report the pre-training modality (Pre-train) and trainable parameters number (Tr. Param.) of each method.



(a) ModelNet40.

| Method | Pre-train | Tr. Param. | Acc.(%) |
|---|---|---|---|
| PointNet++ [49] | N/A | 1.4 M | 90.7 |
| KPConv [64] | N/A | 15.2 M | 92.9 |
| DGCNN [69] | N/A | 1.8 M | 92.9 |
| PointMLP-elite [42] | N/A | 0.68 M | 93.6 |
| PointNeXt [50] | N/A | 1.4 M | 94.0 |
| PointMLP [42] | N/A | 12.6 M | 94.1 |
| RepSurf-U [54] | N/A | 1.5 M | **94.7** |
| DGCNN-OcCo [68] | 3D | 1.8M | 93.0 |
| Point-BERT [81] | 3D | 21.1 M | 93.2 |
| Point-MAE [44] | 3D | 21.1 M | 93.8 |
| P2P (ResNet-101) | 2D | 0.25 M | 93.1 |
| P2P (HorNet-L-22k-mlp) | 2D | 1.2 M | 94.0 |

(b) ScanObjectNN.

| Method | Pre-train | Tr. Param. | Acc.(%) |
|---|---|---|---|
| PointNet++ [49] | N/A | 1.4 M | 77.9 |
| DGCNN [69] | N/A | 1.8 M | 78.1 |
| PRANet [12] | N/A | 2.3 M | 82.1 |
| MVTN [19] | N/A | 14.0 M | 82.8 |
| PointMLP-elite [42] | N/A | 0.68 M | 83.8 |
| PointMLP [42] | N/A | 12.6 M | 85.4 |
| RepSurf-U(2x) [54] | N/A | 6.8 M | 86.1 |
| PointNeXt [50] | N/A | 1.4 M | 88.2 |
| Point-BERT [81] | 3D | 21.1 M | 83.1 |
| Point-MAE [44] | 3D | 21.1 M | 85.2 |
| P2P (ResNet-101) | 2D | 0.25 M | 87.4 |
| P2P (HorNet-L-22k-mlp) | 2D | 1.2 M | **89.3** |



**Advantages of P2P Prompting over Other Tuning Methods.** We conduct extensive ablation studies to demonstrate the advantages of our proposed P2P Prompting over vanilla fine-tuning and other prompting methods, shown in Table 3a. As a baseline (Model A), we directly append classification head to the geometry encoder without the pre-trained image model. Then we incrementally insert pre-trained ViT blocks to process point tokens from the geometry encoder, and discuss different fine-tuning strategies including fixing all ViT weights (Model $B_1$), fine-tuning normalization parameters (Model $B_2$) and fine-tuning all ViT weights(Model $B_3$). We also implement Vision Prompt Tuning (VPT) [24] to Model B with shallow (Model $C_1$) and deep (Model $C_2$) variants.

From the comparisons between Model A and others, we can inspect the contribution of pre-trained knowledge from 2D to 3D classification. However, neither vanilla fine-tuning nor previously prompting mechanism VPT fully exploits the pre-trained image knowledge. Our Point-to-Pixel prompting is the best choice to migrate 2D pre-trained knowledge to 3D domain at a low trainable parameter cost.

**Point-to-Pixel Prompting Designs.** After confirming that P2P is the most suitable tuning mechanism, we discuss the design choices of the P2P module in detail. In Point-to-Pixel Prompting, we produce colorful images to adapt to the pre-trained image model, whose advantages have been discussed in Section 3.3.2. Here we further prove the statement via ablation studies in Table 3b. Model D processes per-pixel feature $\hat{F}$ from Section 3.3.1 to directly generate image tokens and feed them to ViT blocks. In this variant, we bypass the explicit image generation process and directly adopt patch embedding layers on feature map $\hat{F}$. Model E generates binary black-and-white images according to the geometric projection from the point cloud, without predicting pixel colors as in P2P.

According to the results, Model D introduces much more trainable parameters due to the trainable patch embedding projection convolution layer with kernel size 16, while producing inferior classification results than P2P. On the other hand, even though Model E requires fewer trainable parameters, its performance lags far behind. Therefore, producing colorful images as the prompting mechanism can best communicate knowledge between the image domain and point cloud domain, fully exploiting pre-trained image knowledge from the frozen ViT model.

**Influence of Tuning Strategies.** After fixing the architecture of our P2P framework, we investigate the best tuning strategy, adjusting the tuning extent of the pre-trained image model: (1) Model F: training the image model from scratch without loading pre-trained weights. (2) Model G: tuning all ViT parameters. (3) P2P: tuning only normalization parameters. (4) Model H: tuning only bias parameters. (5) Model I: fix all ViT parameters without any tuning.

According to the results in Table 3c, tuning normalization parameters is the most suitable solution, avoiding 2D information lost during massive tuning (model G). Tuning normalization parameters also adapts the model to point cloud data distribution, which model H and I variant fail to accomplish. Additionally, quantitative comparisons between Model F and others demonstrate that the pre-trained knowledge from 2D domain is crucial in our P2P framework, since the limited data in 3D domain is insufficient for optimizing a large ViT model from scratch with numerous trainable parameters.

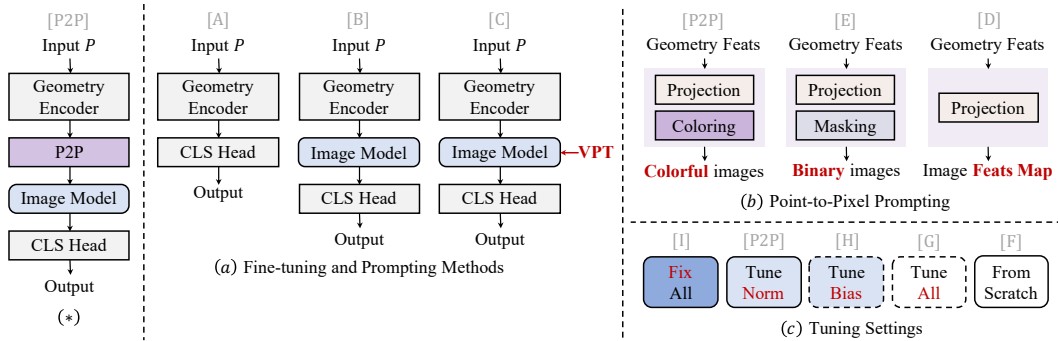

Figure 3: **Ablations illustration.** (∗) shows the pipeline of the overall P2P framework. Part (a) displays ablations on replacing P2P prompting with vanilla fine-tuning or visual prompt tuning (VPT) [24]. Part (b) illustrates ablations on Point-to-Pixel Prompting designs. Part (c) shows different tuning strategies on the pre-trained image model in our P2P framework. Gray letters on top of each model correspond to the Model column in Table 3.

Table 3: **Ablation studies on ModelNet40 classification.** We select ViT-B that is pre-trained on ImageNet-1k as our image model. We report trainable parameters (Tr. Param.) and accuracy (Acc.). (a) shows effects of different tuning strategies, including point-based network without the image model (A), fine-tuning the pre-trained image model to different extents ($B_1$,$B_2$,$B_3$), prompt tuning the pre-trained image model with different variants of VPT ($C_1$,$C_2$) and with our proposed Point-to-Pixel prompting (P2P). (b) shows different Point-to-Pixel Prompting types, discussing whether to explicitly produce images (D) and whether to predict pixel colors (E). (c) shows ablations on tuning settings of the pre-trained image model when training our P2P framework. (d) shows the effect of different pre-training strategies of the image model, where IN Acc. with † and ‡ represent the linear probing and fine-tuning accuracy on ImageNet-1k dataset respectively. * denotes that we implement CoOp to report the zero-shot classification accuracy of the CLIP pre-trained model on ImageNet-1k. Illustrations of ablations (a,b,c) are shown in Figure 3.

(a) Fine-tuning and Prompting Methods.

| Model | Image Model | VPT | P2P | Tr. Param. | Acc.(%) |
|---|---|---|---|---|---|
| A | ✗ | ✗ | ✗ | 7.76 K | 88.5 (-4.2) |
| $B_1$ | Fixed | ✗ | ✗ | 0.08 M | 90.0 (-2.7) |
| $B_2$ | Finetune Norm | ✗ | ✗ | 0.12 M | 90.1 (-2.6) |
| $B_3$ | Finetune All | ✗ | ✗ | 81.2 M | 90.6 (-2.1) |
| $C_1$ | Prompt Tune | Shallow | ✗ | 0.31 M | 90.2 (-2.5) |
| $C_2$ | Prompt Tune | Deep | ✗ | 0.50 M | 90.0 (-2.7) |
| P2P | Prompt Tune | ✗ | ✓ | 0.15 M | 92.7 |

(c) Tuning settings.

| Model | Pre-train | Tune Param. | Tr. Param. | Acc.(%) |
|---|---|---|---|---|
| F | ✗ | All | 81.9 M | 86.3 (-6.4) |
| G | ✓ | All | 81.9 M | 91.7 (-1.0) |
| H | ✓ | Bias | 0.21 M | 92.3 (-0.4) |
| I | ✓ | N/A | 0.11 M | 92.2 (-0.5) |
| P2P | ✓ | Norm | 0.15 M | 92.7 |

(b) Point-to-Pixel Prompting.

| Model | P2P Type | Color | Tr. Param. | Acc.(%) |
|---|---|---|---|---|
| D | Feature | ✗ | 12.1 M | 90.8 (-1.9) |
| E | Image | ✗ | 0.07 M | 89.8 (-2.9) |
| P2P | Image | ✓ | 0.15 M | 92.7 |

(d) Different Pre-training Strategies.

| Model | Pre-train | IN Acc.(%) | Tr. Param. | Acc.(%) |
|---|---|---|---|---|
| J | MAE | 68.0† | 0.15 M | 91.6 |
| K | CLIP | 71.7* | 0.12 M | 91.8 |
| L | MoCo | 76.7† | 0.15 M | 92.3 |
| M | DINO | 78.2† | 0.15 M | 92.8 |
| N | IN 1k | 81.8 | 0.15 M | 92.7 |
| O | IN 22k | 84.0‡ | 0.15 M | 92.9 |

**Effects of Different Pre-training Strategies.** In Table 3d, we show the effects of different strategies for pre-training image models. For supervised pre-training, we load pre-trained weights on ImageNet-1k and ImageNet-22k datasets. For unsupervised pre-training, we select four most representative methods: CLIP [51], DINO [5], MoCo [11] and MAE [20]. We report the linear probing and fine-tuning results on ImageNet-1k dataset of each pre-training strategy in IN Acc. column with † and ‡ respectively. Note that we implement CoOp [86] to report the zero-shot classification accuracy (denoting with ∗) of the CLIP pre-trained model.

From the experiment results, we can conclude that supervised pre-trained image models obtain relatively better results than unsupervised pre-trained ones. This may because the objective of 3D classification is consistent with that in 2D domain, thus the supervised pre-training weight is more suitable to migrate to point cloud classification task. However, unsupervised approach with strong

Table 4: **Part segmentation results on the ShapeNetPart dataset**. We report the mean IoU across all part categories mIoU$_C$ (%) and the mean IoU across all instance mIoU$_I$ (%) , and the IoU (%) for each category.

| Model | mIoU$_C$ | mIoU$_I$ | aero plane | bag | cap | car | chair | ear phone | guitar | knife | lamp | laptop | motor bike | mug | pistol | rocket | skate board | table |
|---|---|---|---|---|---|---|---|---|---|---|---|---|---|---|---|---|---|---|
| PointNet [48] | 80.4 | 83.7 | 83.4 | 78.7 | 82.5 | 74.9 | 89.6 | 73.0 | 91.5 | 85.9 | 80.8 | 95.3 | 65.2 | 93.0 | 81.2 | 57.9 | 72.8 | 80.6 |
| PointNet++ [49] | 81.9 | 85.1 | 82.4 | 79.0 | 87.7 | 77.3 | 90.8 | 71.8 | 91.0 | 85.9 | 83.7 | 95.3 | 71.6 | 94.1 | 81.3 | 58.7 | 76.4 | 82.6 |
| DGCNN [69] | 82.3 | 85.2 | 84.0 | 83.4 | 86.7 | 77.8 | 90.6 | 74.7 | 91.2 | 87.5 | 82.8 | 95.7 | 66.3 | 94.9 | 81.1 | 63.5 | 74.5 | 82.6 |
| Point-BERT [81] | 84.1 | 85.6 | 84.3 | 84.8 | 88.0 | 79.8 | 91.0 | 81.7 | 91.6 | 87.9 | 85.2 | 95.6 | 75.6 | 94.7 | 84.3 | 63.4 | 76.3 | 81.5 |
| PointMLP [42] | 84.6 | 86.1 | 83.5 | 83.4 | 87.5 | 80.5 | 90.3 | 78.2 | 92.2 | 88.1 | 82.6 | 96.2 | 77.5 | 95.8 | 85.4 | 64.6 | 83.3 | 84.3 |
| KPConv [64] | **85.1** | 86.4 | 84.6 | 86.3 | 87.2 | 81.1 | 91.1 | 77.8 | 92.6 | 88.4 | 82.7 | 96.2 | 78.1 | 95.8 | 85.4 | 69.0 | 82.0 | 83.6 |
| P2P (CN-B-SFPN) | 82.5 | 85.7 | 83.2 | 84.1 | 85.9 | 78.0 | 91.0 | 80.2 | 91.7 | 87.2 | 85.4 | 95.4 | 69.6 | 93.5 | 79.4 | 57.0 | 73.0 | 83.6 |
| P2P (CN-L-UPer) | 84.1 | **86.5** | 84.3 | 85.1 | 88.3 | 80.4 | 91.6 | 80.8 | 92.1 | 87.9 | 85.6 | 95.9 | 76.1 | 94.2 | 82.4 | 62.7 | 74.7 | 83.7 |

transferability such as DINO also achieves competitive performance. Secondly, comparing among unsupervised pre-training methods, the one that achieves higher performance with linear probing on 2D classification produces better results in 3D classification. This suggests that the transferability of a pre-trained image model is consistent when migrating to 2D and 3D downstream tasks.

## 4.3  Part Segmentation

The quantitative part segmentation results on ShapeNetPart dataset are shown in Table 4. We implement the base version of ConvNeXt [38] as image model and SemanticFPN [26] as 2D segmentation head for baseline comparison. We further implement the large version of ConvNeXt as the image model and more complex UPerNet [75] as 2D segmentation head to obtain better results. Our P2P framework can achieve better performance than classical point-based methods, which demonstrates its potential in performing 3D dense prediction tasks based on 2D pre-trained image models. We leave it for future work to develop advanced segmentation heads and supervision strategies to better leverage pre-trained 2D knowledge in object-level or even scene-level point cloud segmentation.

## 4.4  Limitations

While P2P shows outstanding classification performance and a promising scaling-up trend, we think that P2P may have difficulty in performing 3D tasks that concentrates on modality-dependent geometry analysis like completion, reconstruction, or upsampling. This is because P2P exploits and transfers the shared visual semantic knowledge between 2D and 3D domains, but these low-level tasks focus more on 3D domain-specific information. Apart from that, even though our P2P framework only requires a few trainable parameters to leverage pre-trained 2D knowledge and obtain high performance, its overall training parameters and FLOPs are still large when the image model is large. We will investigate these problems in future works.

## 5  Conclusion

In this paper, we propose a point-to-pixel prompting method to tune pre-trained image models for point cloud analysis. The pre-trained knowledge in image domain can be efficaciously adapted to 3D tasks at a low trainable parameter cost and achieve competitive performance compared with state-of-the-art point-based methods, mitigating the data-starvation problem in point cloud field that has been an obstacle for massive 3D pre-training researches. The proposed Point-to-Pixel Prompting builds a bridge between 2D and 3D domains, preserving the geometry information of point clouds in projected images while transferring color information from the pre-trained image model back to the colorless point cloud. Experimental results on object classification and part segmentation demonstrate the superiority and potential of our proposed P2P framework.

## Acknowledgments

This work was supported in part by the National Key Research and Development Program of China under Grant 2017YFA0700802, in part by the National Natural Science Foundation of China under Grant 62125603 and Grant U1813218, in part by a grant from the Beijing Academy of Artificial Intelligence (BAAI).

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
