# P2P: Tuning Pre-trained Image Models for Point Cloud Analysis with Point-to-Pixel Prompting
## *Supplemental Material*

**Ziyi Wang**[*]  **Xumin Yu**[*]  **Yongming Rao**[*]
**Jie Zhou**  **Jiwen Lu**[†]
Department of Automation, Tsinghua University, China
{wziyi22, yuxm20}@mails.tsinghua.edu.cn;
raoyongming95@gmail.com;
{lujiwen, jzhou}@tsinghua.edu.cn

## A  More Experimental Results

### A.1  Experiments on Different Pre-trained Image Models

We conduct more experiments on point cloud classification tasks with different image models of different scales, ranging from convolution-based ConvNeXt to attention-based Vision Transformer to Swin Transformer. The image model is pre-trained on ImageNet-22k [1] dataset. We report the image classification performance of the original image model finetuned on ImageNet-1k dataset, the number of trainable parameters after Point-to-Pixel Prompting, and the classification accuracy on ModelNet40 [11] and ScanObjectNN [9] datasets.

From the quantitative results and accuracy curve in Table 1, we can conclude that enlarging the scale of the same image model will result in higher classification performance, which is consistent with the observations in image classification.

### A.2  Ablation Studies on Test View Choices

During training, the rotation angle $\theta$ is randomly selected from $[-\pi, \pi]$ and $\phi$ is randomly selected from $[-0.4\pi, -0.2\pi]$ to keep the objects standing upright in the images. During inference, we evenly divide the range of $\theta$ and $\phi$ into several segments and combine them into multiple views for majority voting. We conduct ablations on the number of views on ModelNet40 dataset with ViT pre-trained on ImageNet-1k dataset as the image model. From the ablation results in Table 2, we choose 10 values of $\theta$ and 4 values of $\phi$ to produce 40 views for majority voting.

### A.3  Ablation Studies on Projection Pooling Strategy

During the geometry-preserved projection, several points may fall in the same pixel. In P2P, we propose to *add* the features of these points altogether for better optimization and keeping geometry density information. Here we conduct ablations on the pooling strategy in Table 3, including max-pooling, mean-pooling and summation. For classification experiment, we report the accuracy on ModelNet40 dataset with ViT-B pre-trained on ImageNet-1k dataset as the image model. For segmentation experiment, we report the instance average IoU on ShapeNetPart dataset with ConvNeXt-B as the image model and SemanticFPN [4] as the segmentation head.

From the classification ablation results, summation is better than max-pooling and mean-pooling. On the one hand, the max-pooling operation drops much geometric information in one pixel. On the

---

[*]Equal contribution.  [†]Corresponding author.

36th Conference on Neural Information Processing Systems (NeurIPS 2022).

Table 1: **More results on ModelNet40 and ScanObjectNN.** We report the image classification performance (IN Acc.) on ImageNet dataset of different image models. After migrating them to point cloud analysis with Point-to-Pixel Prompting, we report the number of trainable parameters (Tr. Param.), performance on ModelNet40 dataset (MN Acc.) and performance on ScanObjectNN dataset (SN Acc.).

(a) Vision Transformer. [2]

| Image Model | IN Acc.(%) | Tr. Param. | MN Acc.(%) | SN Acc.(%) |
|---|---|---|---|---|
| ViT-T | – | 0.10 M | 91.3 | 79.9 |
| ViT-S | – | 0.12 M | 91.9 | 82.6 |
| ViT-B | 84.0 | 0.15 M | 92.4 | 84.1 |
| ViT-L | 85.2 | 0.22 M | 93.2 | 85.0 |

(b) Swin Transformer. [5]

| Image Model | IN Acc.(%) | Tr. Param. | MN Acc.(%) | SN Acc.(%) |
|---|---|---|---|---|
| Swin-T | 80.9 | 0.13 M | 92.5 | 84.2 |
| Swin-S | 83.2 | 0.15 M | 92.8 | 85.6 |
| Swin-B | 85.2 | 0.17 M | 93.2 | 85.8 |
| Swin-L | 86.3 | 0.22 M | 93.4 | 86.7 |

(c) ConvNeXt. [6]

| Image Model | IN Acc.(%) | Tr. Param. | MN Acc.(%) | SN Acc.(%) |
|---|---|---|---|---|
| ConvNeXt-T | 82.9 | 0.12 M | 92.5 | 84.1 |
| ConvNeXt-S | 84.6 | 0.14 M | 92.7 | 86.2 |
| ConvNeXt-B | 85.8 | 0.16 M | 93.2 | 86.5 |
| ConvNeXt-L | 86.6 | 0.19 M | 93.4 | 87.1 |

Table 2: **Ablation studies on test view choices.** We evenly divide $\theta \in [-\pi, \pi]$ and $\phi \in [-0.4\pi, -0.2\pi]$ into multiple segments. We report the classification accuracy on ModelNet40 dataset with ViT-B pre-trained on ImageNet-1k dataset as the image model.

(a) Choices of $\theta$. We choose 4 segments of $\phi$.

| $N_\theta$ | 2 | 4 | 6 | 8 | 10 | 12 |
|---|---|---|---|---|---|---|
| $N_\phi = 4$ | 90.2 | 92.2 | 92.5 | 92.5 | **92.7** | 92.7 |

(b) Choices of $\phi$. We choose 10 segments of $\theta$.

| $N_\phi$ | 2 | 3 | 4 | 5 | 6 |
|---|---|---|---|---|---|
| $N_\theta = 10$ | 92.4 | 92.6 | **92.7** | 92.6 | 92.6 |

other hand, the mean-pooling operation neglects the density information from 3D domain, which also undermines the geometrical knowledge in projected images.

However, in segmentation experiments, the aforementioned three pooling strategies produce the same part segmentation performance. This may be because the multi-hot 2D labels in dense prediction provide extra geometrical guidance that makes up for the gap among different pooling strategies.

### A.4 Visualization of Feature Distributions

Figure 1 shows feature distributions of ModelNet40 and ScanObjectNN datasets in t-SNE visualization. We can conclude that with our proposed Point-to-Pixel Prompting, the pre-trained image model can extract discriminative features from projected colorful images for point cloud analysis.

## B Network Architecture

### B.1 Point-to-Pixel Prompting

The geometry encoder is implemented as a one-layer DGCNN [10] edge convolution. The input points coordinates are first embedded into 8-dim features $F^x$ with a channel-wise convolution. Then we use the k-nearest-neighbor (kNN) algorithm to locate $k = 32$ neighbors $\mathcal{N}_{p_i}$ of each point $p_i$, and concat the central point feature $f_i^x$ with the relative feature $f_j^x - f_i^x$ between each point $p_i$ and neighboring points $p_j \in \mathcal{N}_{p_i}$. Then the concatenated features are processed by a 2D convolution

Table 3: **Ablation studies on projection pooling strategy.** For classification experiment, we report the accuracy on ModelNet40 dataset with ViT-B pre-trained on ImageNet-1k dataset as the image model. For segmentation experiment, we report the instance average IoU on ShapeNetPart dataset with ConvNeXt-B as the image model and SemanticFPN as the segmentation head.

<table>
<tr><td colspan="4">(a) Classification Ablations.</td></tr>
<tr><td>Method</td><td>max</td><td>mean</td><td>sum</td></tr>
<tr><td>Accuracy</td><td>92.2</td><td>92.3</td><td>**92.7**</td></tr>
</table>

<table>
<tr><td colspan="4">(b) Segmentation Ablations.</td></tr>
<tr><td>Method</td><td>max</td><td>mean</td><td>sum</td></tr>
<tr><td>mIoU$_I$</td><td>85.7</td><td>85.7</td><td>85.7</td></tr>
</table>

(a) Feature distribution on ModelNet40.

(b) Feature distribution on ScanObjectNN.

Figure 1: Visualization of feature distributions in t-SNE representations. Best view in colors.

with kernel size 1 followed by a max-pooling layer within all points in $\mathcal{N}_{p_i}$, resulting in a geometry feature $F \in \mathbb{R}^{N \times C}$ of $C = 64$ dims.

In the geometry-preserved projection module, we first calculate the coordinate range $\mathrm{x}^r$ of the input point cloud. Then we calculate the grid size $g_h = H/\mathrm{x}^r, g_w = W/\mathrm{x}^r$ so that the projected object can be fit in the image $I$ with $H = 224, W = 224$.

The coloring module consists of a basic block from ResNet [3] architecture design with $3 \times 3$ convolutions and a final 2D convolution with kernel size 1, smoothing the pixel-level feature distribution and predicting RGB channels of image $I$.

## C   Implementation Details

The implementation details of architectural design and experimental settings are shown in Table 4, where $C_{\mathrm{emb}}$ denotes the embedding dimension of image features extracted by pre-trained image models. We use slightly different architectures for classification and part segmentation. We use 4096 points for ModelNet40 to produce projected images that are relatively smoother, while too few points may lead to sparse and discontinuous pixel distribution in projected images that prevent them from being similar to real 2D images.

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

Table 4: Architecture details and experiment settings of our framework. $C_{\text{emb}}$ denotes the embedding dimension of image features extracted by pre-trained image models.

(a) Architecture of Classification Model.

| Module | Block | $C_{in}$ | $C_{out}$ | Kernel | kNN |
|---|---|---|---|---|---|
| Geometry Encoder | Conv1d | 3 | 8 | 1 | |
| | DGCNN | 8 | 64 | | 32 |
| | Conv1d | 64 | 64 | 1 | |
| Image Coloring | Basic Block | 64 | 64 | 3 | |
| | Conv2d | 64 | 64 | 1 | |
| | Conv2d | 64 | 3 | 1 | |
| CLS Head | Linear | $C_{\text{emb}}$ | 40 | | |

(b) Architecture of Segmentation Model.

| Module | Block | $C_{in}$ | $C_{out}$ | Kernel | kNN |
|---|---|---|---|---|---|
| Geometry Encoder | Conv1d | 3 | 8 | 1 | |
| | DGCNN | 8 | 64 | | 32 |
| | DGCNN | 64 | 128 | | 32 |
| | Conv1d | 128 | 64 | 1 | |
| Image Coloring | Basic Block | 64 | 64 | 3 | |
| | Conv2d | 64 | 64 | 1 | |
| | Conv2d | 64 | 3 | 1 | |
| SEG Head | Semantic FPN | $C_{\text{emb}}$ | 50 | | |

(c) Experiment Settings for Classification.

| Config | Value |
|---|---|
| optimizer | AdamW [8] |
| learning rate | 5e-4 |
| weight decay | 5e-2 |
| learning rate scheduler | cosine [7] |
| training epochs | 300 |
| batch size | 64 |
| GPU device | RTX 3090 Ti |
| image size | 224×224 |
| patch size | 16 |
| drop path rate | 0.1 |
| image normalization | ImageNet style |
| number of points | 4096 (ModelNet)
2048 (ScanObjectNN) |
| augmentation | scale $s \in [2/3, 3/2]$
trans $t \in [-0.2, 0.2]$ |
| rotation angle | $\theta \in [-\pi, \pi]$
$\phi \in [-0.4\pi, -0.2\pi]$ |

[3] Kaiming He, Xiangyu Zhang, Shaoqing Ren, and Jian Sun. Deep residual learning for image recognition. In *CVPR*, 2016.

[4] Alexander Kirillov, Ross Girshick, Kaiming He, and Piotr Dollár. Panoptic feature pyramid networks. In *CVPR*, 2019.

[5] Ze Liu, Han Hu, Yutong Lin, Zhuliang Yao, Zhenda Xie, Yixuan Wei, Jia Ning, Yue Cao, Zheng Zhang, Li Dong, et al. Swin transformer v2: Scaling up capacity and resolution. *arXiv preprint arXiv:2111.09883*, 2021.

[6] Zhuang Liu, Hanzi Mao, Chao-Yuan Wu, Christoph Feichtenhofer, Trevor Darrell, and Saining Xie. A convnet for the 2020s. *arXiv preprint arXiv:2201.03545*, 2022.

[7] Ilya Loshchilov and Frank Hutter. Sgdr: Stochastic gradient descent with warm restarts. *arXiv preprint arXiv:1608.03983*, 2016.

[8] Ilya Loshchilov and Frank Hutter. Decoupled weight decay regularization. *arXiv preprint arXiv:1711.05101*, 2017.

[9] Mikaela Angelina Uy, Quang-Hieu Pham, Binh-Son Hua, Thanh Nguyen, and Sai-Kit Yeung. Revisiting point cloud classification: A new benchmark dataset and classification model on real-world data. In *ICCV*, 2019.

[10] Yue Wang, Yongbin Sun, Ziwei Liu, Sanjay E Sarma, Michael M Bronstein, and Justin M Solomon. Dynamic graph cnn for learning on point clouds. *ToG*, 2019.

[11] Zhirong Wu, Shuran Song, Aditya Khosla, Fisher Yu, Linguang Zhang, Xiaoou Tang, and Jianxiong Xiao. 3d shapenets: A deep representation for volumetric shapes. In *CVPR*, 2015.