# OpenReview forum: "P2P: Tuning Pre-trained Image Models for Point Cloud Analysis with Point-to-Pixel Prompting"
_NeurIPS.cc/2022/Conference — NeurIPS 2022 Accept_

### Official Review · Reviewer_2HDY · 2022-07-04

**Rating:** 6
**Confidence:** 3
**Soundness:** 3 good
**Presentation:** 3 good
**Contribution:** 3 good

**Summary:**

In this paper, the authors propose to leverage the pretrained image model for point cloud downstream tasks. Specifically, they introduce a Point-to-Pixel Prompting to transform a point cloud as the corresponding image, by geometry-preserved projection and geometry-aware coloring.

**Questions:**

See Strengths And Weaknesses

**Limitations:**

See Strengths And Weaknesses

**Strengths And Weaknesses:**

Strengths
1) The paper is well written with clear motivation and good organization.
2) Leveraging 2D pretraining for 3D tasks is an interesting topic.
3) Point-to-Pixel Prompting is novel.

Weakness
1) My main concern is the experiment result. Apparently, the proposed design does not improve the performance.
2) What are the computation cost and model size for the prompting procedure?

---

> ### Author Response · Authors · 2022-08-02
> **Response to Reviewer 2HDY**
>
> Thanks for your careful review and comments! Hopefully the following contents could answer your questions.
>
> ### **1. About the updated experiment results.**
>
> > My main concern is the experiment result. Apparently, the proposed design does not improve the performance.
>
> We implement different image models in Table 1 in our supplementary material, where P2P with ConvNeXt-L as image model achieves 87.1 on ScanObjectNN dataset, surpassing previous literature by a large margin. Sorry for not including it in our main paper.
>
> We further update more comprehensive results in Table 1 in "Response to All Reviewers". From the quantitative results and the accuracy curve, we can conclude that with our proposed P2P prompting method, better 2D pre-trained models in one family will result in better 3D classification performance.
>
> To compare with previous literature, the updated results are shown in Table 2 in "Response to All Reviewers". From the updated results we can conclude that with our proposed P2P prompting method, we achieve the state-of-the-art performance on ScanObjectNN dataset, surpassing previous best works such as PointMLP by a large margin.
>
> For part segmentation experiments, the updated results are shown in Table 3 in "Response to All Reviewers". With ConvNeXt-L as image model and UPerNet as segmentation head, P2P also surpass PointMLP and KPConv on instance mIoU.
>
> Hopefully these updated experiment results will address your concern on P2P performance.
>
> ### **2. About the computation cost and model size for the prompting procedure.**
>
> > What are the computation cost and model size for the prompting procedure?
>
> The FLOPs of the prompting module is 4.2G. The parameters of the prompting module is 81.7k.

---

> > ### Author Response · Authors · 2022-08-06
> > **Looking forward to your feedback**
> >
> > Dear reviewer 2HDY,
> >
> > Does our response address all your concerns? Please feel free to let us know if you have any further questions.
> > To view the comment, please click here: https://openreview.net/forum?id=CZNFw38dDDS&noteId=YZmrhk1BSG
> >
> > Best wishes!

---

### Official Review · Reviewer_xwSJ · 2022-07-10

**Rating:** 4
**Confidence:** 5
**Soundness:** 2 fair
**Presentation:** 3 good
**Contribution:** 3 good

**Summary:**

The paper proposes point-to-pixel prompting to leverage 2D pre-trained models to help 3D point cloud recognition tasks. The main modules include a geometry-preserved projection and a geometry-aware coloring, which fill the gap between 3D point clouds and 2D images. The experiments on ModelNet40 and ScanObjectNN show that P2P achieves comparable performance on classification tasks with only a few trainable parameters.

**Questions:**

Please refer to weaknesses. Some more questions are listed below:
1. For the projection from 3D to 2D, which are the pixel features for those pixels without points?
2. How to obtain point predictions for part segmentation is unclear. It seems not very reasonable to simply add the multi-view predictions. Actually, the multi-view fusion is not clearly stated in the paper.
3. How's the processing of the empty pixels in the coloring module? The visualization results look very clean and not very smooth actually, and I wonder if the empty pixels are filtered out in the coloring module.

**Limitations:**

The limitation is discussed in Sec 4.3.

**Strengths And Weaknesses:**

Strengths:
1. The paper first proposes a prompt-tuning method to adopt 2D pre-trained parameters in 3D, which is an interesting and novel exploration.
2. With p2p prompting, the model can achieve competitive results on the shape classification task with much fewer trainable parameters.

Weaknesses:
1. Although the method leverages extra 2D image knowledge, it does not show clear performance or speed advantages over previous 3D networks on both classification and part segmentation. The parameters that need to be trained are fewer but the whole model is larger. The 2D prior knowledge is not fully exploited in this method.
2. The design of simply adding the point features in the same pixel seems trivial, and even with the explanations in Line190-197, I don't really think it preserves geometry. Also, no more experiments are conducted to analyze these design choices.
3. More results on scene-level point cloud understanding with datasets like ScanNet or S3DIS are expected to illustrate the effectiveness of the prompt-tuning pipeline.

---

> ### Author Response · Authors · 2022-08-02
> **Response to Reviewer xwSJ (1/2)**
>
> Thanks for your careful review and comments! Hopefully the following contents could answer your questions.
>
> ### **1. About the experiment results.**
>
> > Although the method leverages extra 2D image knowledge, it does not show clear performance or speed advantages over previous 3D networks on both classification and part segmentation. The parameters that need to be trained are fewer but the whole model is larger. The 2D prior knowledge is not fully exploited in this method.
>
> We implement different image models in Table 1 in our submitted supplementary material. Sorry for not including it in our main paper. We further update more comprehensive results in Table 1 in "Response to All Reviewers". From the quantitative results and the accuracy curve, we can conclude that with our proposed P2P prompting method, larger 2D pre-trained models in one family will result in better 3D classification performance. We hope that this scalable trend will address your concern on how much we exploit the 2D pre-trained knowledge.
>
> To compare with previous literature, the updated results are shown in Table 2 in "Response to All Reviewers". From the updated results we can conclude that with our proposed P2P prompting method, we achieve the state-of-the-art performance on ScanObjectNN dataset, surpassing previous best works such as PointMLP by a large margin. For part segmentation experiments, the updated results are shown in Table 3 in "Response to All Reviewers". With ConvNeXt-L as image model and UPerNet as segmentation head, P2P also surpass PointMLP and KPConv on instance mIoU.  Hopefully these updated experiment results will address your concern on P2P performance.
>
> ### **2. About the projection process.**
>
> #### **(A) Adding features for points in one pixel.**
>
> > The design of simply adding the point features in the same pixel seems trivial, and even with the explanations in Line190-197, I don't really think it preserves geometry. Also, no more experiments are conducted to analyze these design choices.
>
> Thanks for pointing out the lack of ablation on how to aggregate features of multiple points in one pixel.  We conduct ablations on max-pooling, taking average and summation, shown in the following table. We implement ViT-B as our image model on ModelNet40 dataset, which is pre-trained on ImageNet-1k dataset with supervised classification.
>
> | Method    | Accuracy |
> | :-------: | :------: |
> | max       | 92.2     |
> | mean      | 92.3     |
> | sum       | 92.7     |
>
> As shown in Table 4, the quantitative results show that the summation design is the best choice. What's more, according to the visualization results of projected and colored images in Figure 1 in our paper, the objects appear to be semi-transparent, which to some extent demonstrates the preservation of geometrical information from 3D point clouds in 2D images.
>
> #### **(B) Features for pixels without points.**
>
> > For the projection from 3D to 2D, which are the pixel features for those pixels without points?
>
> Sorry for the unclear statement. Features for pixels without points are initialized as zeros.
>
> #### **(C) Processing of the empty pixels in the coloring module.**
>
> > How's the processing of the empty pixels in the coloring module? The visualization results look very clean and not very smooth actually, and I wonder if the empty pixels are filtered out in the coloring module.
>
> Features for pixels without points are initialized as zeros. The convolution layers in the coloring module are applied to the whole projected image to predict color for each pixel. We don't explicitly filter out empty pixels in the coloring module, as learnable bias parameters in the convolution layers would predict the same color (greenish gray in our visualization) for pixels with zero value. As for the smoothing problem, it is caused by the sparsity of the point cloud. We try to solve it by the convolution layers in the coloring module with $3\times 3$ kernel size.

---

> > ### Author Response · Authors · 2022-08-02
> > **Response to Reviewer xwSJ (2/2)**
> >
> > ### **3. About multi-view fusion in part segmentation.**
> >
> > > How to obtain point predictions for part segmentation is unclear. It seems not very reasonable to simply add the multi-view predictions. Actually, the multi-view fusion is not clearly stated in the paper.
> >
> > Sorry for the unclear statement about the multi-view fusion. During evaluation, we re-project pixel-level predictions back to points according to Point-to-Pixel projection correspondences. Segmentation probabilities for each point from multiple views are added together for majority voting. More specifically, if we denote $c^k_i \in \mathcal{R}^{N}$ as the segmentation prediction of point $i$ from view $k$, where $N$ is the number of total classes. Then the added prediction for totally $K$ views would be $c_i=\sum^K_{k=1}c^k_i$. The final part segmentation result for point $i$ would be $n=\textrm{arg}\max_N c_i$.
> >
> > The reason why we use a multi-view prediction summation is that one pixel may correspond to multiple points from one projection view. This will cause two problems. Firstly, point-wise segmentation boundaries are blurred, since multiple points in a local region are predicted to be the same class. Secondly, segmentation confidence for points would be less distinguishable. For example, if three points $p_1, p_2, p_3$ belonged to different classes $c_1, c_2, c_3$ are projected in the same pixel $i_a$ and the multi-hot segmentation confidence for $i_a$ is $[1/3, 1/3, 1/3, 0]$, then the argmax operation cannot decide which class $p_1, p_2, p_3$ belong to. However, suppose that from another view, $p_1, p_4$ belonged to $c_1, c_4$ are projected to the same pixel $i_b$ with segmentation confidence $[1/2, 0, 0, 1/2]$, then the segmentation probability for $p_1$ would be $[5/6, 1/3, 1/3, 1/2]$. Under this condition, the argmax operation could correctly predict that $p_1$ belongs to $c_1$.
> >
> > ### **4. About scene level point cloud understanding tasks.**
> >
> > > More results on scene-level point cloud understanding with datasets like ScanNet or S3DIS are expected to illustrate the effectiveness of the prompt-tuning pipeline.
> >
> > Thanks for your constructive suggestion. However, our main concern in this paper is utilizing object-level experiments to demonstrate that migrating pre-trained knowledge from 2D domain to 3D tasks is a novel and feasible learning paradigm for 3D development. We also show the potential of P2P in dense prediction tasks with experiments on part segmentation. For more complex scene level detection and segmentation, we hope we can study them more thoroughly in future work.

---

> > > ### Author Response · Authors · 2022-08-06
> > > **Looking forward to your feedback**
> > >
> > > Dear reviewer xwSJ,
> > >
> > > Does our response address all your concerns? Please feel free to let us know if you have any further questions.
> > > To view the comment, please click here: https://openreview.net/forum?id=CZNFw38dDDS&noteId=3IImbi3g0GB
> > >
> > > Best wishes!

---

### Official Review · Reviewer_xwQ5 · 2022-07-10

**Rating:** 6
**Confidence:** 4
**Soundness:** 3 good
**Presentation:** 4 excellent
**Contribution:** 3 good

**Summary:**

In this paper, the authors introduce point-to-pixel prompting (P2P), a learning framework for leveraging pre-trained image transformers for 3D tasks. The method is mainly motivated by the data scarcity issue in 3D domains. P2P learns a geometry-preserving transformation from point cloud to 2D grid, and then a projection to prepare the 2D grid data to be processed by a pre-trained image transformer expecting image tokens. The main benefit of P2P, to my understanding, is the ability to achieve comparable accuracy to other 3D models with much fewer parameters that need to be trained with 3D data. This is validated on two tasks, 3D object classification and 3D part segmentation.

**Questions:**

Instead of taking a sum of point features in each cell (L189), did you try max pooling? It would be great to see a sensitivity analysis comparing {max,mean,sum} pooling for this, since images of semi-transparent objects (which is unrealistic) would seemingly be out-of-distribution for pre-trained image models on ImageNet?

Currently, the strengths and weaknesses of the paper roughly balance each other out in my opinion and I believe the work as it stands is borderline. I would be interested to read the authors response to my feedback. Thanks!

============================
After reading the author's responses to my questions and concerns I have increased my score to reflect that I feel the strengths now outweigh the weaknesses.



**Limitations:**

No. Limitations of the P2P framework should be discussed in the main text (e.g., in the conclusions section).

**Strengths And Weaknesses:**

Strengths

-----------------
- the question of whether knowledge can be transferred from large pre-trained image models for use with 3D domains is interesting
- the point-to-pixel prompt pipeline, which is nicely visualized in Figure 2, appears to be novel and is simple and elegant
- this work is a nice demonstration of ideas from NLP transferring successfully to other domains (in this case, to 3D point cloud processing)
- the paper is well-written and easy to read


Weaknesses

-----------------
- $\text{\textbf{Unclear motivation,problem, and significance}}$: The current set of claims in the introduction are that A) there is a data starvation problem in 3D domain (L34-35) and B) pre-training point cloud transformers suffers from an imbalance between the number of trainable parameters and limited training data, leading to insufficient optimization and overfitting (L40-41). However, the data starvation problem seems to only exist for specific object-centric datasets such as ShapeNet. By contrast, consider the large Scannet and Waymo datasets. Moreover, recent advances in 3D rendering (e.g., NeRF) suggests that highly lifelike synthetic 3D data may soon become available. Therefore, scarcity of large datasets does not appear to be a fundamental concern. Moreover, point B) seems plainly false since recent methods like Point-BERT work just as well as P2P on, e.g., ModelNet40.
- As a result, it is unclear what the actual problem is that is being addressed here and *why* this prompting method is needed at all. The main benefit of P2P seems to be in the use of fewer model parameters, but its unclear why this is important.
- $\text{\textbf{Multiple unsubstantiated claims}}$. These can be addressed with careful editing.
  - (L54) “The end-to-end optimization pipeline and the strategy of freezing the pre-trained image model promote the *bidirectional* knowledge flow between points and pixels”. To my understanding, the flow is *unidirectional*; pre-trained image features are being used to learn a better representation for points.
  - (L271) “Firstly, our P2P outperforms traditional 3D pretraining methods” (on ModelNet40). P2P’s largest model achieves the same performance as Point-BERT.
  - Similarly, claims of “superiority” of P2P (L64, L370) are clearly not supported by the accuracy results in the experiments.


References

-----------------
- Dai, Angela, Angel X. Chang, Manolis Savva, Maciej Halber, Thomas Funkhouser, and Matthias Nießner. "Scannet: Richly-annotated 3d reconstructions of indoor scenes." In Proceedings of the IEEE conference on computer vision and pattern recognition, pp. 5828-5839. 2017.
- Sun, Pei, Henrik Kretzschmar, Xerxes Dotiwalla, Aurelien Chouard, Vijaysai Patnaik, Paul Tsui, James Guo et al. "Scalability in perception for autonomous driving: Waymo open dataset." In Proceedings of the IEEE/CVF conference on computer vision and pattern recognition, pp. 2446-2454.

---

> ### Author Response · Authors · 2022-08-02
> **Response to Reviewer xwQ5 (1/2)**
>
> Thanks for your careful review and detailed comments! Hopefully the following contents could answer your questions.
>
> ### **1. About the unclear motivation, problem and significance.**
>
> #### **(A) The data starvation problem.**
>
> > However, the data starvation problem seems to only exist for specific object-centric datasets such as ShapeNet. By contrast, consider the large Scannet and Waymo datasets. Moreover, recent advances in 3D rendering (e.g., NeRF) suggests that highly lifelike synthetic 3D data may soon become available. Therefore, scarcity of large datasets does not appear to be a fundamental concern.
>
> Sorry for the misunderstanding caused by our motivation claims. We agree that there are large-scale scene-level datasets ScanNet and Waymo, and that recent advances in 3D rendering is promising to produce more synthetic 3D data. However, our main concern is that the scale and generalizability of these 3D datasets are relatively weaker than their counterparts in 2D domain. For example, there are ImageNet-1k that contains 1.2M images from 1000 categories, not to mention the larger ImageNet-21k dataset. After all, it is much easier to obtain various images from the Internet. On the contrary, there are only 1513 scenes containing 20 categories in indoor dataset ScanNet, while outdoor dataset Waymo also contains no more than 30 categories. Therefore, their diversity and volume lag behind 2D pre-training datasets. Another advantage of 2D pre-training is that it can consistently scale-up. In other words, larger model size and larger data size will consistently produce higher performance. Numerous mature pre-training methods based on ImageNet are proposed based on this property and show promising performances on both classification and downstream tasks.
>
> Therefore, it would be great if the abundant datasets and outstanding pre-training mechanism in 2D domain could help 3D development, since they both illustrate the visual world and share many similarities. And this doesn't contradict with more data in 3D domain. Our main contribution is proposing a new learning paradigm to leverage 2D pre-training knowledge to 3D domain at a low trainable parameter cost. We implement different image models and the results are included in our submitted supplementary material. We further update more comprehensive results in Table 1 in "Response to All Reviewers". From the quantitative results and the accuracy curve, we can conclude that with our proposed P2P prompting method, the scaling-up property in 2D domain is successfully kept, as larger 2D pre-trained models in one family will result in better 3D classification performance. This demonstrates the feasibility of leveraging the abundant 2D datasets and the development in 2D pre-training to 3D domain.
>
> #### **(B) Results comparisons with pre-trained point cloud Transformers.**
>
> > Moreover, point B) seems plainly false since recent methods like Point-BERT work just as well as P2P on, e.g., ModelNet40.
>
> To compare with previous literature, the updated results are shown in Table 2 in "Response to All Reviewers". From the updated results we can conclude that with our proposed P2P prompting method, we achieve the state-of-the-art performance on ScanObjectNN dataset, surpassing previous best works such as Point-BERT, PointMLP by a large margin. On ModelNet40 dataset, we also surpass Point-BERT and PointMLP-elite.
>
> For part segmentation experiments, the updated results are shown in Table 3 in "Response to All Reviewers". With ConvNeXt-L as image model and UPerNet as segmentation head, P2P also surpass Point-BERT, PointMLP and KPConv on instance mIoU. Hopefully these updated experiment results will address your concern on P2P performance.
>
> #### **(C) Benefits of the proposed P2P prompting.**
>
> > As a result, it is unclear what the actual problem is that is being addressed here and why this prompting method is needed at all. The main benefit of P2P seems to be in the use of fewer model parameters, but its unclear why this is important.
>
> There are three benefits of our proposed P2P. **Firstly, high performance.** According to Table 2, our P2P obtain state-of-the-art performance on ScanObjectNN dataset and surpass previous literature by a large margin. **Secondly, consistent scaling-up trend.** According to Table 1, our proposed prompting mechanism could largely benefit the remarkable progress in 2D pre-training, as larger scale image model in one family will consistently result in better 3D performance. **Thirdly, low prompting cost.** Prompting is an important mechanism to transfer pre-trained knowledge to downstream tasks at a low tuning cost. It would benefit from recent advances in fundamental models and would contribute to the future unified model researches, as different input modalities and different output tasks can share the same large-scale fundamental model and only require light-weight prompting module for adaptation.

---

> > ### Author Response · Authors · 2022-08-02
> > **Response to Reviewer xwQ5 (2/2)**
> >
> > ### **2. About the flow between pixel and points.**
> >
> > > To my understanding, the flow is unidirectional; pre-trained image features are being used to learn a better representation for points.
> >
> > Sorry for the unclear statement in the introduction. We further discuss the bidirectional knowledge flow in Section 3.1 (L141-149 in original paper, L119-127 in revised paper). The flow from point to pixel is more direct than the opposite: the output color of each pixel is influenced by the point features, since the pixel features are obtained from point features according to the projection correspondences. Therefore, pixel colors embrace geometry information from point clouds.
> >
> > ### **3. About the result comparison with Point-BERT.**
> >
> > > P2P’s largest model achieves the same performance as Point-BERT.
> >
> > > Similarly, claims of “superiority” of P2P (L64, L370) are clearly not supported by the accuracy results in the experiments.
> >
> > Sorry for the claims that are not that rigorous in the first version. However, with our updated experiment results, our P2P obtains state-of-the-art performance on ScanObjectNN dataset and surpass previous literature by a large margin, as shown in Table 2. For ModelNet40 dataset, we surpass Point-BERT and PointMLP-elite.
> >
> > ### **4. About the ablation on pooling strategy in the projection process.**
> >
> > > Instead of taking a sum of point features in each cell (L189), did you try max pooling? It would be great to see a sensitivity analysis comparing {max,mean,sum} pooling for this.
> >
> > Thanks for pointing out the lack of ablation studies on point feature aggregation in each pixel. The ablation on max/mean/sum pooling are shown in the following table. We implement ViT-B as our image model on ModelNet40 dataset, which is pre-trained on ImageNet-1k dataset with supervised classification.
> >
> > | Method    | Accuracy |
> > | :-------: | :------: |
> > | max       | 92.2     |
> > | mean      | 92.3     |
> > | sum       | 92.7     |
> >
> > According to the results, summation operation is better than max pooling or mean pooling, which is consistent with what we have discussed in Section 3.2.2 (L189-197 in original paper, L167-175 in revised paper). On the one hand, the max pooling operation drops much geometric information in one pixel. On the other hand, the mean pooling operation neglect the density information from 3D domain, which also undermines the geometrical knowledge in projected images.
> >
> > > ..., since images of semi-transparent objects (which is unrealistic) would seemingly be out-of-distribution for pre-trained image models on ImageNet?
> >
> > As for the out-of-distribution problem, we agree that the semi-transparent objects are not similar in **texture** as objects in realistic images. However, the **shape** information of object are hardly affected by the semi-transparent attribute. Given that the image model relies on both shape and texture for classification[1], domain gap in texture is a trade-off choice as it won't be a decisive factor. What's more, we try to solve this problem by tuning the normalization layers of the image model. There are two supporting reasons. Firstly, the normalization parameters will affect the image texture style according to StyleGAN[2]. Secondly, some early literature such as AdaBN[3] proposed to match normalization parameters for domain adaptation.
> >
> > ### **References**
> >
> > [1] Tuli, Shikhar, et al. "Are Convolutional Neural Networks or Transformers more like human vision?." arXiv preprint arXiv:2105.07197. 2021.
> > [2] Karras, Tero, Samuli Laine, and Timo Aila. "A style-based generator architecture for generative adversarial networks." CVPR. 2019.
> > [3] Li, Yanghao, et al. "Adaptive batch normalization for practical domain adaptation." Pattern Recognition 80 (2018): 109-117.

---

> > > ### Comment · Reviewer_xwQ5 · 2022-08-05
> > > **Response to author**
> > >
> > > Hi,
> > >
> > > Thanks so much for your responses to my comments and questions.
> > > - The explanation about the data starvation problem (the diversity of categories of large scale 3D datasets being limited compared to ImageNet-1k and 21k) makes sense. Thanks.
> > > - One follow-up question I have is whether there is an intuitive explanation behind why P2P with ResNet-101 on ScanObjectNN has the best performance, despite ResNet-101 having relatively lower accuracy (IN acc. = 77.4 in Table 1a). This result seems somewhat surprising and would benefit from an explanation?
> > > - I don't agree with the conclusion about the pooling strategy ablation that it definitively shows that geometry information is preserved by the sum pooling point feature aggregation step. The difference between the three pooling strategies seems negligible. It seems more likely that the considered task with the largest demonstrated improvement (object classification) doesn't need fine-grained geometric features. Can you think of any 3D point cloud processing tasks which are more dependent on geometric information than object classification which P2P might not show improvements on? It would be helpful to have a discussion about this when discussing limitations of P2P (which are still missing from the paper...)
> > > - Are there any other limitations of P2P? (Will a discussion of this be added to a new revision of the paper?)

---

> > > > ### Author Response · Authors · 2022-08-05
> > > > **Response to Reviewer xwQ5**
> > > >
> > > > Thank you so much for your response to our comments! We wish the following response could address your concerns.
> > > >
> > > > #### **1. About the results of P2P with ResNet-101 on ScanObjectNN.**
> > > >
> > > > The reason is that the transferability of different image models to 3D domain are different. But we can see that the scaling-up trend of the transferability **within the same architecture** is consistent. The scaling-up trend in 2D domain is the property that a larger model and more data are guaranteed to produce higher performance. This is important since it can leverage the development in hardware and datasets. As our P2P also shows such property in 3D domain, we are optimistic that with our P2P framework, 3D point cloud analysis can still benefit from future development in 2D pre-training models. We will include this analysis in our revised paper.
> > > >
> > > > #### **2. About the pooling strategy ablation.**
> > > >
> > > > We agree that fine-grained geometric features in classification are not much important compared with other dense prediction tasks, as the final prediction is dependent on a single global feature. However, recent work like RepSurf[1] finds that the encoding of detailed local geometry can bring a positive effect on 3D models. But we still agree that pooling strategy ablations on dense prediction tasks like part segmentation that requires per-point comprehension will be more powerful and convincing than ablations on classification. We will conduct additional experiments on part-segmentation to further verify the conclusion we get in the last response and include it in our revised paper. Thanks for your suggestion!
> > > >
> > > > #### **3. About the limitation of P2P.**
> > > >
> > > > Thanks for your advice! We will add a more thorough limitation discussion of P2P in our revised paper. We think that P2P may have difficulty in performing 3D tasks that concentrates on modality-dependent geometry analysis like completion, reconstruction, or upsampling. This is because P2P exploits and transfers the shared visual semantic knowledge between 2D and 3D domains, but these low-level tasks focus more on 3D domain-specific information. Apart from that, even though our P2P framework only requires a few trainable parameters to leverage pre-trained 2D knowledge and obtain high performance, its overall training parameters and FLOPs are still large when the image model is large. We will investigate this problem in future works.
> > > >
> > > > ### Reference
> > > > [1] Ran, Haoxi, Jun Liu, and Chengjie Wang. "Surface Representation for Point Clouds." CVPR. 2022.

---

> > > > > ### Comment · Reviewer_xwQ5 · 2022-08-06
> > > > > **Response**
> > > > >
> > > > > Thanks for your responses, which I believe are reasonable. Considering that these important discussions will be included in the final camera ready version (there is an extra page for such additions) I have increased my score to 6.

---

> > > > > > ### Author Response · Authors · 2022-08-06
> > > > > > **Thanks for upgrading your score and providing valuable feedback.**
> > > > > >
> > > > > > Thanks for upgrading your score and providing valuable feedback. We will update our revised paper according to our discussions. Thank you again for your insightful and constructive suggestions that improve paper quality!

---

### Official Review · Reviewer_RwM1 · 2022-07-13

**Rating:** 6
**Confidence:** 2
**Soundness:** 3 good
**Presentation:** 3 good
**Contribution:** 3 good

**Summary:**

This paper proposes a new model architecture for 3D problem which leverages the powerful backbones pretrained from the 2D task. The idea is straightforward. The input point cloud is projected into 2D pixels using an encoder model, then the 2D pixels is colored by a coloring module, and the colored images are fed into a pretrained ViT backbone and then predictions are made by the task-specific heads. The overall approach provides an elegant solution to leverage the representations of 2D models. The experimental results demonstrate superior performance on public benchmarks including ModelNet40, ShapeNetPart datasets.

**Questions:**

How much improvements are from the improved backbone architecture and added compute cost? It would be great to show an ablation study on that.

**Ethics Review Area:**

["I don’t know"]

**Limitations:**

The proposed approach has some limitations on the feasible tasks to apply on. For example, it may not work if we want to conduct a 3D segmentation task.
Related to the question above, the comparison in Table 4 is not fair due to lack of model complexity analysis.

**Strengths And Weaknesses:**

1. Novel model architectures to utilize pretrained 2d models. To my knowledge, the idea of using projection into 2D and coloring module is new.
2. The idea is simple yet effective. The pretrained 2d models are easy to get and the results are promising.

---

> ### Author Response · Authors · 2022-08-02
> **Response to Reviewer RwM1**
>
> Thanks for your careful review and comments! Hopefully the following contents could answer your questions.
>
> ### **1. About the ablations on improved backbone architecture and added compute cost.**
>
> > How much improvements are from the improved backbone architecture and added compute cost? It would be great to show an ablation study on that.
>
> Thanks for your insightful suggestion about ablations on improved backbone architecture. Actually, we've conducted similar ablation studies on image models pre-trained on ImageNet-21k[6] dataset in Table 1 in our submitted supplementary material. Sorry for not including it in our main paper. We further update more comprehensive results in Table 1 in "Response to All Reviewers". From the quantitative results, we can conclude that with our proposed P2P prompting method, larger 2D pre-trained models from one family will result in better 3D classification performance.
>
> As for the added computation cost, since we use the same prompting module for different image model, the FLOPs from P2P prompting remains the same: **4.2G**. The FLOPs of the improved image models are referenced from their original papers and shown in the following table.
>
> | ResNet    | FLOPs | ViT   | FLOPs | Swin | FLOPs | ConvNeXt | FLOPs |
> | :-------: | :---: | :---: | :---: | :--: | :---: | :------: | :---: |
> | ResNet-18 | 1.8G  | ViT-T | 1.1G | Swin-T | 4.5G | ConvNeXt-T | 4.5G |
> | ResNet-50 | 3.8G  | ViT-S | 4.6G | Swin-S | 8.7G | ConvNeXt-S | 8.7G |
> | ResNet-101 | 7.6G  | ViT-B | 17.5G | Swin-B | 15.4G | ConvNeXt-B | 15.4G |
> | | | | | | | ConvNeXt-L | 34.4G
>
> ### **2. About the potential of P2P on segmentation task.**
>
> > The proposed approach has some limitations on the feasible tasks to apply on. For example, it may not work if we want to conduct a 3D segmentation task.
>
> To illustrate the potential of P2P to be applied on tasks other than classification, we conduct experiments on part segmentation and our improved results with improved image model are shown in Table 3 in "Response to All Reviewers". The part segmentation results on instance mIoU surpass the previous outstanding works like KPConv, which demonstrates the potential of our P2P to perform segmentation tasks.
>
> > Related to the question above, the comparison in Table 4 is not fair due to lack of model complexity analysis.
>
> Thanks for pointing out the lack of complexity analysis in Table 4. The trainable parameters for each model is listed as below.
>
> | Model | Trainable Parameters   |
> | :---- | :--------------------: |
> | PointNet++            | 1.4 M  |
> | DGCNN                 | 1.8 M  |
> | Point-BERT            | 21.1 M |
> | PointMLP              | 12.6 M |
> | KPConv                | 15.2 M |
> | *P2P (ViT-B-MAE-SFPN, original)* | 0.3 M |
> | P2P (ConvNeXt-B-SFPN) | 6.1 M  |
> | P2P (ConvNeXt-L-UPer) | 71.7 M |
>
> With ConvNeXt-B-SFPN setting under a relatively low trainable parameter cost, we achieve competitive performance. With a stronger ConvNeXt-L-UPer setting, we use more parameters to obtain higher segmentation performance. The extra trainable parameters mainly come from the UPerNet segmentation head. We agree that the trainable parameters for ConvNeXt-L-UPer setting is relatively large, but we want to emphasize that the scaling-up porperty of our proposed P2P framework is crucial. In future works, we plan to analyze P2P in segmentation more thoroughly for better performance-parameter balance.

---

> > ### Author Response · Authors · 2022-08-06
> > **Looking forward to your feedback**
> >
> > Dear reviewer RwM1,
> >
> > Does our response address all your concerns? Please feel free to let us know if you have any further questions.
> > To view the comment, please click here: https://openreview.net/forum?id=CZNFw38dDDS&noteId=GoAQDbVo5CE
> >
> > Best wishes!

---

### Author Response · Authors · 2022-08-02
**Response to All Reviwers (1/2)**

We would like to thank all reviewers for their careful review and insightful feedback! We are excited that they found our proposed idea to be "novel"[R1,R2,R3,R4] and "interesting"[R2,R3,R4] and our proposed framework to be "elegant"[R1,R2].

We also appreciate their suggestions to make our work better. We notice that many reviewers have concerns on the experiment results. To further demonstrate the effectiveness of our proposed P2P framework, we've updated some experiment results as listed below. We also update these experiment results in our revised paper, where our major revisions are marked blue.

### **1. P2P variants with different image models.**
Thanks to Reviewer RwM1's suggestion, we implement different scales of convolution-based ResNet[1], ConvNeXt[2] and attention-based Vision Transformer[3], Swin Transformer[4] as the image model in our P2P framework. These image models are pre-trained on ImageNet-1k[5] dataset with supervised classification. We report classification accuracy on ModelNet40 (MN Acc.) and ScanObjectNN (SN Acc.) datasets. We also report the classification accuracy of the image model on ImageNet (IN Acc.). We report trainable parameters of each P2P framwork with Tr. Param. Note that we've conducted similar ablation studies on image models pre-trained on ImageNet-21k[6] dataset in Table 1 in our submitted **supplementary material**. Here we make this ablation more comprehensive. This table is corresponding to Table 1 in our revised paper.
#### **Table 1. Classification Results of P2P Variants with Different Image Models.**
#### *(a) ResNet.*
| Image Model | IN Acc. | Tr. Param. | MN Acc. | SN Acc. |
| :---------- | :----:  | :--------: | :-----: | :-----: |
| ResNet-18   | 69.8    | 109 K      | 91.6    | 82.6    |
| ResNet-50   | 76.1    | 206 K      | 92.5    | 85.8    |
| ResNet-101  | 77.4    | 257 K      | 93.1    | 87.4    |
#### *(b) Vision Transformer.*
| Image Model | IN Acc. | Tr. Param. | MN Acc. | SN Acc. |
| :---------- | :----:  | :--------: | :-----: | :-----: |
| ViT-T       | 72.2    | 99 K       | 91.5    | 79.7    |
| ViT-S       | 79.8    | 116 K      | 91.8    | 81.6    |
| ViT-B       | 81.8    | 150 K      | 92.7    | 83.4    |
#### *(c) Swin Transformer.*
| Image Model | IN Acc. | Tr. Param. | MN Acc. | SN Acc. |
| :---------- | :----:  | :--------: | :-----: | :-----: |
| Swin-T      | 81.3    | 136 K      | 92.1    | 82.9    |
| Swin-S      | 83.0    | 154 K      | 92.5    | 83.8    |
| Swin-B      | 83.5    | 178 K      | 92.6    | 84.6    |
#### *(d) ConvNeXt.*
| Image Model | IN Acc. | Tr. Param. | MN Acc. | SN Acc. |
| :---------- | :----:  | :--------: | :-----: | :-----: |
| ConvNeXt-T  | 82.1    | 126 K      | 92.6    | 84.9    |
| ConvNeXt-S  | 83.1    | 140 K      | 92.8    | 85.3    |
| ConvNeXt-B  | 83.8    | 159 K      | 93.0    | 85.7    |
| ConvNeXt-L  | 84.3    | 198 K      | 93.2    | 86.2    |

### **2. Classification results comparisons with previous literature.**
When comparing with previous literature, we show a base version and an advanced version of P2P. In the base version, we implement ResNet-101 as image model and use a simple fully connected layer as classification head. In the advanced version, we implement ConvNeXt-L pre-trained on ImageNet-21k[6] dataset as our image model and replace the fc classification head by a multi-layer perceptron (MLP). We also show the results of P2P(ViT-B-MAE) setting in our original paper. This table is corresponding to Table 2 in our revised paper. We report the classification accuracy on ModelNet40 (MN) and ScanObjectNN (SN). We also report the trainable parameters with Tr. Param. and pre-training type for each method.
#### **Table 2. Classification Results on ModelNet40 and ScanObjectNN**
| Method      | Pre-train | Tr. Param. | MN Acc. | SN Acc. |
| :---------- | :------:  | ---------: | :-----: | :-----: |
| PointNet++  | N/A       | 1.4 M      | 90.7    | 77.9    |
| DGCNN       | N/A       | 1.8 M      | 92.9    | 78.1    |
| MVTN        | N/A       | 14.0 M     | N/A     | 82.8    |
| PointMLP-elite | N/A    | 0.68 M     | 93.6    | 83.8    |
| PointMLP    | N/A       | 12.6 M     | 94.1    | 85.4    |
| DGCNN-OcCo  | 3D        | 1.8M       | 93.0    | N/A     |
| Point-BERT  | 3D        | 21.1 M     | 93.2    | 83.1    |
| *P2P (ViT-B-MAE, original)* | 2D   | 0.17 M      | 93.2    | 84.5    |
| P2P (ResNet-101) | 2D   | 0.25 M     | 93.1    | 87.4    |
| P2P (ConvNeXt-L-21k-mlp) | 2D | 1.0 M  | 93.7    | 87.6   |

---

> ### Author Response · Authors · 2022-08-02
> **Response to All Reviwers (2/2)**
>
>
> ### **3. Segmentation results comparisons with previous literature.**
> We show the updated part segmentation results as below. The base version uses a ConvNeXt-B as image model with SemanticFPN[7] as segmentation head. The advanced version uses a ConvNeXt-L as image model with UPerNet[8] as segmentation head. We also show the results of P2P(ViT-B-MAE-SFPN) setting in our original paper. This table is corresponding to Table 4 in our revised paper. We report the mean IoU across all part
> categories with mIoU_C and the mean IoU across all instance with mIoU_I.
> #### **Table 3. Part Segmentation Results**
> | Model       | mIoU_C | mIoU_I |
> | :---------- | :----: | :----: |
> | DGCNN       | 82.3   | 85.2   |
> | Point-BERT  | 84.1   | 85.6   |
> | PointMLP    | 84.6   | 86.1   |
> | KPConv      | 85.1   | 86.4   |
> | *P2P (ViT-B-MAE-SFPN, original)* | 81.7   | 85.0   |
> | P2P (ConvNeXt-B-SFPN) | 82.5   | 85.7   |
> | P2P (ConvNeXt-L-UPer) | 84.1   | 86.5   |
>
> ### **References**
> [1] He, Kaiming, et al. "Deep residual learning for image recognition." CVPR. 2016.
> [2] Liu, Zhuang, et al. "A convnet for the 2020s." CVPR. 2022.
> [3] Dosovitskiy, Alexey, et al. "An image is worth 16x16 words: Transformers for image recognition at scale." arXiv preprint arXiv:2010.11929. 2020.
> [4] Liu, Ze, et al. "Swin transformer: Hierarchical vision transformer using shifted windows." ICCV. 2021.
> [5] Krizhevsky, Alex, Ilya Sutskever, and Geoffrey E. Hinton. "Imagenet classification with deep convolutional neural networks." NeurIPS. 2012.
> [6] Ridnik, Tal, et al. "Imagenet-21k pretraining for the masses." arXiv preprint arXiv:2104.10972. 2021.
> [7] Kirillov, Alexander, et al. "Panoptic feature pyramid networks." CVPR. 2019.
> [8] Xiao, Tete, et al. "Unified perceptual parsing for scene understanding." ECCV. 2018.

---

### Meta-Review · Area_Chair_7EAb · 2022-08-25

**Recommendation:** Accept
**Confidence:** Certain

**Metareview:**

The paper presents a method of prompt tuning to transfer 2D pre-trained weights to tackling 3D understanding problems. All reviewers are positive about the novelty of the method. With large 2D pretrained models, higher performances are still expected from xwSJ, which is also a reasonable comment. Other 3D understanding tasks, such as segmentation and detection of outdoor scenes, are strongly encouraged, as they are the true needs of the industry.

**Award:**

No

---

### Decision · Program_Chairs · 2022-09-14

Accept